# Phylogenetic and Transcriptional Analyses of the *HSP20* Gene Family in Peach Revealed That *PpHSP20-32* Is Involved in Plant Height and Heat Tolerance

**DOI:** 10.3390/ijms231810849

**Published:** 2022-09-16

**Authors:** Xiaodong Lian, Qiuping Wang, Tianhao Li, Hongzhu Gao, Huannan Li, Xianbo Zheng, Xiaobei Wang, Haipeng Zhang, Jun Cheng, Wei Wang, Xia Ye, Jidong Li, Bin Tan, Jiancan Feng

**Affiliations:** 1College of Horticulture, Henan Agricultural University, Zhengzhou 450002, China; 2Henan Engineering Center for Peach Germplasm Innovation and Utilization, Zhengzhou 450002, China; 3College of Forestry, Henan Agricultural University, Zhengzhou 450002, China

**Keywords:** peach (*Prunus persica*), HSP20, functional identification, PpHSP20-32, thermotolerance

## Abstract

The heat shock protein 20 (HSP20) proteins comprise an ancient, diverse, and crucial family of proteins that exists in all organisms. As a family, the HSP20s play an obvious role in thermotolerance, but little is known about their molecular functions in addition to heat acclimation. In this study, 42 *PpHSP20* genes were detected in the peach genome and were randomly distributed onto the eight chromosomes. The primary modes of gene duplication of the PpHSP20s were dispersed gene duplication (DSD) and tandem duplication (TD). PpHSP20s in the same class shared similar motifs. Based on phylogenetic analysis of HSP20s in peach, *Arabidopsis thaliana*, *Glycine max,* and *Oryza sativa*, the PpHSP20s were classified into 11 subclasses, except for two unclassified PpHSP20s. *cis*-elements related to stress and hormone responses were detected in the promoter regions of most PpHSP20s. Gene expression analysis of 42 *PpHSP20* genes revealed that the expression pattern of *PpHSP20-32* was highly consistent with shoot length changes in the cultivar ‘Zhongyoutao 14’, which is a temperature-sensitive semi-dwarf. *PpHSP20-32* was selected for further functional analysis. The plant heights of three transgenic Arabidopsis lines overexpressing PpHSP20-32 were significantly higher than WT, although there was no significant difference in the number of nodes. In addition, the seeds of three over-expressing lines of PpHSP20-32 treated with high temperature showed enhanced thermotolerance. These results provide a foundation for the functional characterization of *PpHSP20* genes and their potential use in the growth and development of peach.

## 1. Introduction

During these times of global environmental change, temperature is arguably the most important factor affecting plant growth and geographical distribution [1,2]. Plants experience fluctuations in the thermal environment characterized by average, maximum, and minimum daily temperatures, which change over the course of the seasons. Plants are very sensitive to temperature, showing responses to slight changes of just 1 °C [3]. However, it remains unknown how temperature signals are perceived. Extremely high or low temperatures lead to temperature stress, which is one of the most severe abiotic stressors and severely impacts plant growth [4,5,6]. Plants have evolved multiple pathways to adapt to temperature stress [7,8,9,10]. Nevertheless, the precise mechanism of temperature sensing, particularly ambient temperature, is still unclear.

Tremendous amounts of research have shown that multiple heat shock proteins (HSPs) emerge as central players in the temperature response, including the growth/development and stress responses [11,12,13,14]. According to their molecular weight, HSPs are divided into five major categories, including *HSP100s*, *HSP90s*, *HSP70s*, *HSP60s*, and *HSP20s* [4]. Among the five major categories, HSP20s are the most prevalent in plants. HSP20s are also known as small heat shock proteins and have molecular weights ranging from 15 to 42 kDa. Structurally, HSP20s are conserved, with the alpha-crystallin domain (ACD) in the C-terminus [15,16]. HSP20s have been identified in a few plant species, such as Arabidopsis (*Arabidopsis thaliana*) [17], rice (*Oryza sativa*) [18], pepper (*Capsicum annuum*) [19], tomato (*Solanum lycopersicum*) [20], watermelon (*Citrullus lanatus*) [21], grape (*Vitis vinifera*) [22], *Sorghum bicolor* [23], and apple (*Malus domestica*) [13].

*HSP20s* participate in a wide range of developmental processes and abiotic stresses, such as heat, salt, and drought [19,24,25,26]. In tomatoes, four *SlHSP20s* are constitutively expressed in almost all tissues, indicating that they might play specific housekeeping functions under normal growth conditions [20]. Under heat, drought, and salt stresses, the expression levels of *SlHSP20s* were up-regulated, implying their potential roles in regulating the response to stresses [20]. Similar results were obtained in peppers, where most *CaHSP20s* were highly induced by heat stress [19]. Overexpression of *CaHSP16.4* enhanced the tolerance to heat and drought stresses by stabilizing the ROS-scavenging system [27]. Unlike heat stress, few *HSP20s* were induced by cold stress in soybean [28]. In grapes, most *VvHSP20s* responded to H_2_O_2_ treatment [22]. In addition, plenty of HSP20s were related to plant development, including seed maturation and germination, pollen development, and hypocotyl elongation [16]. *AtHSP22* participated in auxin-regulated hypocotyl elongation under high temperatures in *Arabidopsis thaliana* [26].

Peach (*Prunus persica* L.), an economically important crop and a popular fruit with consumers, is widespread in both temperate and subtropical regions, although peach trees can also be found in high altitude and severe cold regions [29]. Due to its wide distribution, genetic diversity, and relatively small genome size, the peach is considered a model species for genomic research of rosaceae [30]. ‘Zhongyoutao 14’, a temperature-sensitive semi-dwarf (*TSSD*) peach cultivar, showed extremely shortened internodes below 30 °C and normal internode length above 30 °C [10,31]. Due to shorter internodes until mid-May and then normal length internode in the orchard of the Zhengzhou Fruit Research Institute, the tree heights of *TSSD* categorize them as semi-dwarf [31]. Whether *PpHSP20* participated in the temperature sensitivity of ‘Zhongyoutao 14’ is unknown. In this study, the expression patterns of *PpHSP20s* were analyzed during shoot elongation of ‘Zhongyoutao 14’ at four critical growth stages. Furthermore, the function of *PpHSP20-32* was analyzed via ectopic expression in transgenic Arabidopsis plants. The thermotolerant characteristics of transgenic Arabidopsis carrying *PpHSP20-32* were also analyzed. This study provides valuable information as a step toward the further investigation of the functions and regulatory mechanisms of potentially important *PpHSP20* genes that may be crucial in temperature tolerance and plant height in peach trees.

## 2. Results

### 2.1. Whole Genome Identification and Protein Structure of Peach HSP20 Genes

Forty-two *HSP20* gene family members were identified from the peach genome and named *PpHSP20-1* to *PpHSP20-42*, according to their physical location on the peach chromosomes (Table 1 and Appendix A). The physicochemical properties of the predicted PpHSP20 proteins were analyzed. The predicted molecular weight ranged from 15.5 kDa to 100.0 kDa, except for PpHSP20-18 and PpHSP20-30, which were less than 15 kDa. The number of amino acids ranged from 87 aa (PpHSP20-30) to 919 aa (PpHSP20-42), and the predicted isoelectric point ranged from 4.59 (PpHSP20-42) to 9.44 (PpHSP20-36). Most PpHSP20 were predicted to be unstable proteins, while a small portion (19.0%) were stable, with instability index values less than 40 (which is a stable protein). The average hydrophobic index values of all family members were negative, indicating that the PpHSP20 protein family members were hydrophobic proteins.

The predicted protein secondary structure showed that all PpHSP20s contained four secondary structures. The proteins were mainly composed of alpha helix and random coil motifs, while beta turns were the least identified structures. Except for PpHSP20-34, PpHSP20-36, and PpHSP20-40, most of the PpHSP20s (71.4% of 42 PpHSP20s) contained second structures in the ranked quantities of random coils > alpha helix ≥ extended strand > beta turn, while the quantity ranking of the structures of the other members (21.4% of 42 PpHSP20s) was random coil > extended chain structure > alpha helix > beta turn (Table 2).

### 2.2. Chromosome Distribution, Gene Duplication, Gene Structure, and Conserved Motif Analysis of PpHSP20 Genes

The *PpHSP20* genes were unevenly distributed among the eight chromosomes of peach (Figure 1). Among them, chromosomes 1 and 7 had five *PpHSP20s*, and chromosomes 2, 3, and 8 each had four *PpHSP20s*. Chromosome 4 carried three *PpHSP20* genes, chromosome 5 had six, and Chromosome 6 had the most *PpHSP20s*, at eleven.

The expansion of the PpHSP20 genes family in peach was analyzed (Figure 1). The main expansion patterns were dispersed gene duplication (DSD; for 18 or 42.9% of the *PpHSP20* genes) and tandem duplication (TD; for 13 *PpHSP20* genes or 30.9%), (Figure 1 and Appendix A). Eight genes arose through whole-genome duplication (WGD), including PpHSP20-7, PpHSP20-8, PpHSP20-17, PpHSP20-24, PpHSP20-25, PpHSP20-32, PpHSP20-38, and PpHSP20-42) (Figure 1).

The 42 PpHSP20s were grouped into five classes (Figure 2A), containing 20, 4, 2, 3, and 13 PpHSP20s in Class I–V, respectively. The structural differences of the *PpHSP20* genes were also predicted. Fourteen *PpHSP20s* were intronless (33.3%), 17 (40.5%) had one intron, and seven (16.7%) had two introns (Figure 2B). The remaining four *PpHSP20* genes contained more than two introns: *PpHSP20-13* had five introns, *PpHSP20-17* had 14 introns, *PpHSP20-38* had 11 introns, and *PpHSP20-42* had 12 introns.

MEME was used to analyze the conserved motifs of the PpHSP20 proteins, and the results showed that PpHSP20s contained ten conserved motifs. Among all PpHSP20 members, 40 (95.2%) contained Motif 1, 20 (47.6%) contained Motif 2, 34 (81.0%) contained Motif 3, and 32 (76.2%) contained Motif 6. Motif analysis showed that the PpHSP20s containing similar motifs were grouped in the same class (Figure 2A,C). For example, most PpHSP20s in class I contained motifs 1, 2, 3, 5, 6, and 10. Motifs 1, 2, 3, 6, 8, and 10 were contained in the HSP20 domain (Figure 2C).

### 2.3. Phylogenetic Analysis of PpHSP20 Proteins

A phylogenetic analysis was conducted using 128 HSP20 proteins, including 19 *Arabidopsis thaliana* sequences, 45 *Glycine max* sequences, 22 *Oryza sativa* sequences, and 42 peach sequences (Figure 3). The 128 HSP20s were divided into 13 different subfamilies, including CI, CII, CIII, CIV, CV, CVI, CVII (cytoplasm or nucleus), MI, MII (mitochondria), ER (endoplasmic reticulum), P (Plastid), Po (Peroxisome), and an unknown classification (Figure 3). There were 43 CIs, 13 CIIs, 21 CIIIs, 6 CIVs, 5 CVs, 3 CVIs, 1 CVIIs, 4 MIs, 5 MIIs, 11 Ps, 5 Pos, 9 ERs, and 2 in unknown classification. Most of the PpHSP20s (30, 71.4%) were classified as CI–CVI, followed by P (2), and ER (2); moreover, MI, MII, and Po contained only one PpHSP20 each. However, PpHSP20-32 and PpHSP20-41 were not classified. This indicated that the PpHSP20s likely function in the cytoplasm or nucleus, while a few were distributed in organelles.

### 2.4. Analysis of Cis-Acting Elements of PpHSP20s Promoters

The promoters in the upstream 2000 bp region of 42 *PpHSP20* genes were analyzed to identify the *cis*-elements. Eleven types of *cis*-elements were detected. Most of the *PpHSP20* genes possessed abscisic acid-responsive (ABRE), light-responsive, MeJA-responsive, and anaerobic-induction elements (Figure 4A,B). The elements were grouped into three categories, including phytohormone-responsive, abiotic, and biotic stress-responsive and plant development-related *cis*-elements. The phytohormone-responsive classification accounted for the highest proportion (49.2%, 186 of 378 elements), including abscisic acid-responsive), MeJA-responsive, salicylic acid-responsive (SA), auxin-responsive and gibberellin-responsive (GA). The abiotic and biotic stress-responsive elements included anaerobic induction and low temperature-responsive elements. In the plant development-related category, seed specific regulation, cell cycle regulation, and circadian control were identified. These results suggested that PpHSP20s were not only related to stress response, but also related to other physiological responses.

### 2.5. Expression of PpHSP20s during the Shoot Elongation of ‘Zhongyoutao 14’

The expression patterns of the *PpHSP20s* were compared at four critical stages (initial period, IP; initial elongation period, IEP; rapid growth period, RGP; stable growth period, SGP) of shoot elongation in the temperature-sensitive semi-dwarf peach cultivar ‘Zhongyoutao 14’, grown in the field under regular management with natural ambient temperature. According to their expression patterns, the 42 *PpHSP20s* could be classified into four groups (Figure 5A). Group I contained 19 *PpHSP20s* that showed the highest expression level during SGP. Group II contained 7 *PpHSP20s* that showed the lowest expression level in IP, including *PpHSP20-6*, *PpHSP20-17*, *PpHSP20-18*, *PpHSP20-21*, *PpHSP20-33*, *PpHSP20-37,* and *PpHSP20-38*. The transcript abundance of *PpHSP20s* in Group III was higher in the IP and the IEP (*PpHSP20-32*, *PpHSP20-13*, *PpHSP20-14*, and *PpHSP20-35*), compared to RGP and the SGP. Group IV, the second largest group containing 12 PpHSP20s, showed higher expression levels in IP or RGP, compared to IEP and RGP. The internodes length of IP (1.21 mm) and IEP (2.57 mm) were significantly less than that of RGP (11.27 mm) and SGP (12.54 mm) [10]. It showed a negative trend between the internode length and the expression levels of *PpHSP20s* in Group III (marked in red). Among the Group III genes, the expression level of *PpHSP20-32* was most consistent with the terminal internode length, as shown in Figure 5B. We speculated that *PpHSP20-32* might participate in temperature-induced shoot growth in this temperature-sensitive peach cultivar.

### 2.6. Overexpression of PpHSP20-32 in Arabidopsis Leads to an Increase in Plant Height

In order to study the function of PpHSP20-32, we constructed a *PpHSP20-32* overexpression vector and transformed it into *Arabidopsis thaliana* using an *Agrobacterium*-mediated method. The phenotypes of three transgenic lines (L1, L2, and L3) and WT were recorded (Figure 6). Two weeks after being transplanted into a substrate, their rosette leaves were longer and wider than WT (average length and width), but there was no significant difference in the number of rosette leaves (Figure 6A–C). Four weeks after transplanting, the plant morphology was observed, and the height of the flowering bolt in the three transgenic lines was higher than that of WT (Figure 6D,E). The average plant height of the three transgenic lines was 34.0 cm (L1, 34.4 cm; L2, 34.3 cm; L3, 33.4 cm), which was significantly higher than the 29.3 cm of WT. The average lengths of the internodes of the transgenic line 1–3 and WT were 1.67 cm, 1.89 cm, 1.63 cm, and 1.83 cm, respectively (Figure 6F), showing no significant difference between the transgenic lines and WT. The average numbers of internodes of the transgenic line 1–3 and WT were 23, 22, 20, and 17, respectively (Figure 6G). There was also no significant difference in the number of branches among all lines (Figure 6H).

### 2.7. PpHSP20-32-OE Seeds Exhibit Enhanced Thermotolerance

The seeds of three PpHSP20-32-OE lines and WT were treated at 46 °C for 30 min and transferred to 25 °C to assay thermotolerance (Figure 7). By 48 h after heat stress (HS), there was no seed germination in any of the four lines (Figure 7A,B). After 60 h at high temperature, the germination rate of the three PpHSP20-32 transgenic lines was 100%, which was significantly higher than that of WT seeds (Figure 7C,D,F). For the WT, the germination was less than 10% after 60 h of HS (Figure 7C,F), but reached about 100% germination at 96 h (Figure 7E,F). These results suggested that the overexpression of *PpHSP20-32* improves the resistance of Arabidopsis seeds to high temperatures.

## 3. Discussion

As plants sense high temperatures or heat stress, gene expression patterns will vary, especially the up-regulation of the heat shock genes [11,32]. HSPs include the HSP100s, HSP90s, HSP70s, HSP60s, and HSP20s. HSP20s are a diverse, ancient, and important family among the HSPs [16]. The number of HSP20s has been determined in numerous plants, such as 31 in *Arabidopsis thaliana* [17,33], 51 in *Glycine max* [28], 35 in *Capsicum annuum* [19], 42 in *Solanum lycopersicum* [20], 63 in *Panicum virgatum* [34], 48 in *Solanum tuberosum* [35], 48 in *Vitis vinifera* [22], 47 in *Sorghum bicolor* [23], 41 in *Malus pumila* [13], 47 in *Cucumis sativus* [14], 45 in *Cucumis melo* [14], and 47 in *Citrullus lanatus* [14]. In peach, we identified 42 *PpHSP20s*, a number greater than in *Arabidopsis thaliana* and pepper, but lower than in switchgrass, potato, and grape. The varied numbers in different plants might be due to the difference in gene duplications during the evolution of the plants.

The 42 *PpHSP20s* are unevenly mapped on the eight chromosomes, with Chr6, the second longest chromosome, containing the most HSP20s. The members of other gene families, such as E3 genes, were also mainly mapped on the longer chromosomes in peach [36], while the F-box genes showed a similar phenomenon in pear [37]. The E3 and F-box genes were mapped on the longer chromosome, similar to the distribution of PpHSP20s on the chromosomes. However, the biggest cluster of HSP20s was on the shortest chromosome, chromosome 8, in apple [13]. So, any rules of distribution of gene family members may be different among different families or different plants, and need to be further validated.

The *PpHSP20* duplication in peach showed inconsistent patterns with those of other plants [13,28,35]. In apple, WGD and TD were the main duplication patterns [13]. In this study, the *PpHSP20* family expanded more by DSD and TD. In peach, DSD was the major expansion route for other gene families, such as the F-box, U-box, RING, BTB, SKP [36], and HSF genes [38]. This phenomenon in peach might be explained by the fact that the peach genome has not undergone a recent whole-genome duplication [39].

The 42 PpHSP20s were divided into 11 classes (CI, CII, CIII, CIV, CV, CVI, MI, MII, P, ER, and Px), except for 2 unclassified PpHSP20s, based on the phylogenetic tree which was constructed using the amino acid sequences of peach, rice, *Arabidopsis thaliana* and soybean. In an earlier study, the AtHSP20s were divided into seven classes (CI, CII, CIII, M, P, ER, and Px), except for five genes that did not fall into any class [17]. Afterwards, the five unclassified AtHSP20s were categorized into five new classes (CIV, CV, CVI, and CVII) and MII [40]. Most of PpHSP20s were classified into nucleocytoplasmic subfamilies (CI–CVI), which indicated that the cytoplasm may be the primary site of action for the HSP20 proteins. This was also observed in other plants, for example, apple and soybean [13,28]. In this study, the HSP20s of peach lacked any proteins in the CVII subgroup, similar to soybean [28], rice [18], switchgrass [34], apple [13], and three cucurbit species (cucumber, melon, and watermelon) [14]. In *Arabidopsis thaliana*, the CVII subgroup gene *AtHsp14.7* was constitutively expressed, and its transcript level did not change under different stresses [40]. It was speculated that AtHsp14.7-CVII was involved in specific housekeeping functions [40].

Plant HSPs are molecular chaperones that protect the functions of their target proteins under various stress conditions to help maintain growth and development [4,16]. In this study, PpHSP20s showed different expression patterns at non-stressful but elevated temperature. The expression patterns of four PpHSP20s, namely *PpHSP20-13*, *PpHSP20-14*, *PpHSP20-35*, and especially *PpHSP20-32*, showed a correlation with the length of the terminal internodes in the shoots of temperature-sensitive semi-dwarf peach. *Populus trichocarpa*, a transgenic line overexpressing *PtHSP17.8*, showed enhanced tolerance to heat and salt stresses [24]. In pepper, *CaHSP16.4* participated in heat and drought stress by enhancing the scavenging of reactive oxygen species [27]. These results mainly focused on the function of HSP20s under stress conditions. Based on this study, *PpHSP20s* might play important role in the regulation of shoot elongation at non-stressful temperatures. In addition, HSP20 responded to the phytohormone ABA and modulated polar auxin transport [26].

In the ambient temperature-sensing pathway, *AtHSP70* is expressed at a level proportionate to the ambient temperature [41]. *AtHSP90* integrates environmental temperature and auxin signaling to regulate temperature-dependent plant growth by stabilizing the auxin co-receptor F-box protein TIR1 [42,43]. A recent study showed that the heat shock protein *AtHSP22* promoted hypocotyl elongation at high temperatures by regulating polar auxin transport, which required the ABI1 protein phosphatase [26]. In this study, *PpHSP20-32*-overexpressing transgenic lines produced larger rosette leaves and taller plants than WT. The plant height of the transgenic lines was higher than that of WT. There was no significant difference in the length of the internodes between the transgenic lines and WT, indicating that the increase in plant height of the transgenic lines may be caused by the increase in the number of internodes. The *PpHSP20-32*-overexpressing lines also showed enhanced heat tolerance. Similar results were observed in rice, pepper, and poplar, which together demonstrate that HSP20 genes enhance thermotolerance [24,25,27].

It remains unknown how PpHSP20-32 regulates rosette leaf size and plant height. The promoter of *PpHSP20-32* contained four types of phytohormone-responsive elements, namely ABRE, MeJA-responsive, salicylic acid-responsive, and gibberellin-responsive elements (Figure 4A,B). This indicated that *PpHSP20-32* might serve as a phytohormone responsive factor. In *Arabidopsis thaliana*, *AtHSP22* is regulated by ABA and auxin, while *AtHSP22* potentiates the auxin efflux PIN proteins, which promotes hypocotyl elongation [26]. These results suggested that PpHSP20-32 might serve to modulate rosette leaf and flower bolt growth by orchestrating phytohormone signaling.

## 4. Materials and Methods

### 4.1. Plant Materials

The peach cultivar ‘Zhongyoutao14’, a temperature-sensitive semi-dwarf, is planted in the experimental station of the Horticulture College, Henan Agricultural University (Zhengzhou, China). The shoot internode length was temperature-dependent. Shoot tips were collected at four critical growth stages, namely the initial period (IP), initial elongation period (IEP), rapid growth period (RGP), and stable growth period (SGP) [10]. The internodes’ lengths were less than 3 mm at IP and IEP with lower environmental temperature (below 30 °C). While the internodes’ lengths at RGP and SGP with higher temperatures (above 30 °C) were more than 10 mm [10]. All samples were quickly frozen in liquid nitrogen after collection and stored in the laboratory at −80 °C. *Arabidopsis thaliana* (L.) Heynh Columbia 0 (Col-0) was used for transformation with PpHSP20-32.

### 4.2. Identification and Characteristic Analysis of Peach HSP20s

The hidden Markov model (HMM) profile (PF00011), characteristic of HSP20, was downloaded from the Pfam website (http://pfam.xfam.org, accessed on 10 May 2022) and used to identify HSP20 genes in peach. An hmmsearch was performed against the peach genome files (v2.1), downloaded from the JGI database (https://phytozome.jgi.doe.gov/pz/portal.html, accessed on 10 May 2022). The isoelectric points and other physical properties were approximated from ExPASy (http://web.expasy.org/compute_pi, accessed on 10 May 2022).

### 4.3. Chromosome Location and Gene Structure Analysis of the PpHSP20 Genes

According to the genome location annotation given by Phytozome V12.1, the chromosome location of each *PpHSP20* was mapped using TBtools [44]. According to the DNA and CDS sequences data for the peach HSP20 gene, the gene structure map was drawn using the online tool GSDS (http://gsds.cbi.pku.edu.cn/, accessed on 10 May 2022).

### 4.4. Phylogeny and Motif Analysis of PpHSP20s

The amino acid sequences of the HSP20 genes of *Arabidopsis thaliana*, *Oryza sativa*, *Glycine max,* and peach were saved as FASTA format files. The phylogenetic tree was constructed by the maximum likelihood method using MEGA 7.0 software (v7.0, Sudhir kumar, Hachioji, Tokyo, Japan) [45]. The online software MEME5.0.4 (http://alternate.meme-suite.org/tools/meme, accessed on 12 May 2022) was used to analyze the motifs in each protein sequence.

### 4.5. Analysis of Cis-Acting Elements of PpHSP20s

The promoter sequence of each *PpHSP20* gene (2000 bps upstream of the start codons) was downloaded from the peach genome. The *cis*-acting elements of the HSP20 promoters were detected using PlantCARE (http://bioinformatics.psb.ugent.be/webtools/plantcare/html/, accessed on 10 May 2022).

### 4.6. PpHSP20 Gene Expression in Different Growth Stages of Peach

The expression levels of the *PpHSP20* genes were obtained from our previous transcriptome data of the four critical stages of shoot growth in the cultivar ‘Zhongyoutao 14’ (Appendix A) [10]. The heatmap was generated by TBtools (v1.09876, Chengjie Chen, Guangzhou, Guangdong, China) [44]. The FPKM (fragments per kilobase of exon per million fragments mapped) values of the HSP20s and the terminal internode lengths of the stems were used for the correlation analysis.

### 4.7. Generation and Phenotypic Observation of PpHSP20-32-Overexpression in Arabidopsis Lines

The CDS of *PpHSF20-32* was amplified using PpHSP20-32-F and PpHSP20-32-R primers (Appendix A). The resulting *PpHSP20-32* fragment was inserted into the pMD19T vector (Takara, Dalian, China). After sequence confirmation, the full coding sequence of *PpHSP20-32* was amplified with primers including restriction sites *Hind* III and *Xba* I (Appendix A), and the amplified fragment was directionally inserted into the vector pSAK277. Transgenic Arabidopsis plants were generated through the floral dip method using the *Agrobacterium tumefaciens* strain GV3101 [46].

After screening for kanamycin resistance and PCR verification (an initial denaturing step at 98 °C for 5 min, followed by 30 cycles of 98 °C for 10 s, 55 °C for 15 s, and 72 °C for 40 s, then 72 °C for 3 min), the transgenic plants were allowed to flower. Seeds from T_2_ transgenic Arabidopsis lines were used for subsequent experiments. Three seedlings from each line with five rosette leaves per seedling of each line was considered one biological replicate and used for the observation of leaf phenotype (length and width). Three biological replicates were taken, for a total of nine seedlings observed. The height, the length of internodes, and the number of internodes and branches (five plants per line) in the different transgenic lines and WT were determined.

### 4.8. Heat Stress Treatment

To detect the function of PpHSP20-32 in heat tolerance, heat stress treatment (46 °C for 30 min) was performed. Before heat stress treatment, seeds of WT and transgenic Arabidopsis lines were sown on MS medium and kept dark at 4 °C for 2 d, and then at 22 °C for 2 d. Then, the seeds were exposed to 46 °C for 30 min, followed by being transferred into a climate chamber (22 °C, 16 h light/8 h dark cycles). The germination of seeds was counted daily and photographed. More than 60 seeds of each line were used in each plate, with three replicates. Differences in heat stress tolerance were confirmed using Student’s *t*-test.

### 4.9. Statistical Analysis

Data were analyzed by ANOVA, Tukey HSDa, and Duncan’s multiple range tests (at *p* < 0.05) using IBM SPSS Statistics 20 (SPSS, Armonk, New York, NY, USA).

## 5. Conclusions

In this study, 42 *PpHSP20* genes, distributed on eight chromosomes randomly, were identified in the peach genome. Dispersed gene duplication (DSD) and tandem duplication (TD) were the primary modes of gene duplication of PpHSP20s. Except for two unclassified *PpHSP20s*, the other 40 *PpHSP20s* were classified into 11 subclasses. The gene structures, basic classification, conserved motifs, and *cis*-elements were also analyzed. The expression pattern of *PpHSP20-32* was highly consistent with shoot length changes during four critical growth stages of temperature-sensitive semi-dwarf peach ‘Zhongyoutao 14’ in response to increasing temperature. Transgenic Arabidopsis lines overexpressing *PpHSP20-32* demonstrated that the gene can increase plant height and enhance thermotolerance. The results in this study supplied general information on the PpHSP20 gene family and further revealed the potential roles of PpHSP20-32 in plant height, in addition to the response to heat stress.

## Figures and Tables

**Figure 1 ijms-23-10849-f001:**
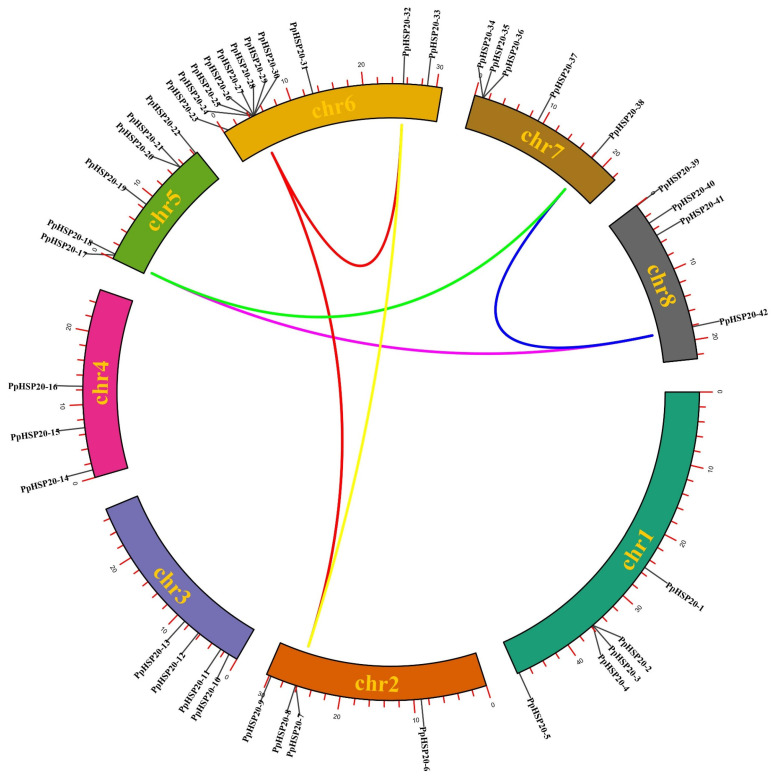
Genomic distribution and duplication of the *PpHSP20* genes across the eight chromosomes of peach. Forty-two *PpHSP20* genes were mapped to the eight chromosomes. Syntenic pairs are linked with lines, with colors representing different pairs.

**Figure 2 ijms-23-10849-f002:**
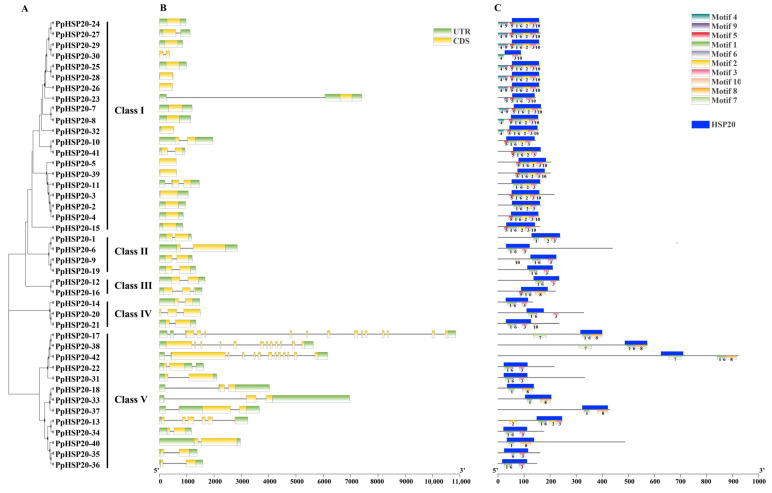
Phylogenetic tree of the PpHSP20 genes (**A**), gene structures (**B**), and conserved motifs of the PpHSP20s (**C**).

**Figure 3 ijms-23-10849-f003:**
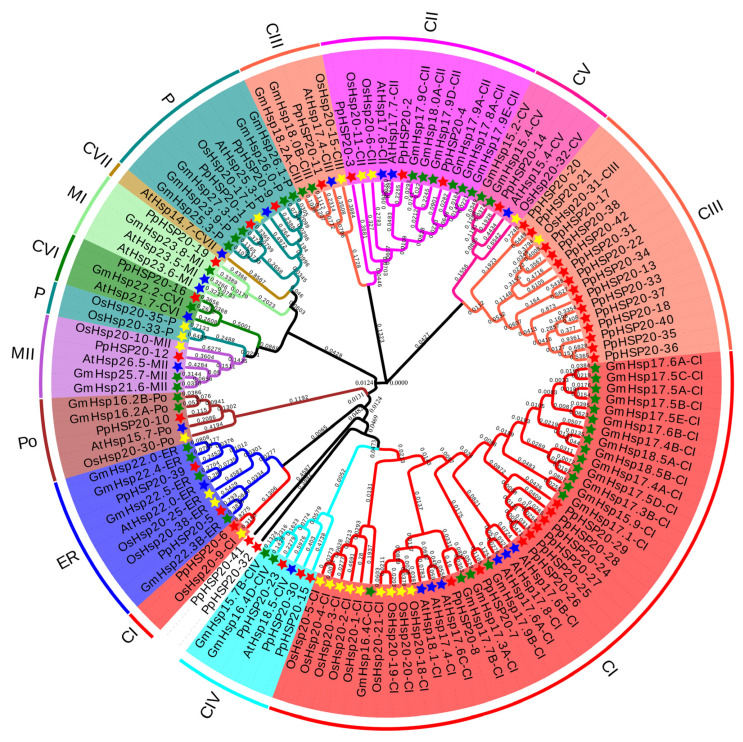
Phylogenetic tree of HSP20s from *Prunus persica* (red star), *Oryza sativa* (yellow star), *Glycine max* (green star), and *Arabidopsis thaliana* (blue star) constructed by the neighbor-joining method.

**Figure 4 ijms-23-10849-f004:**
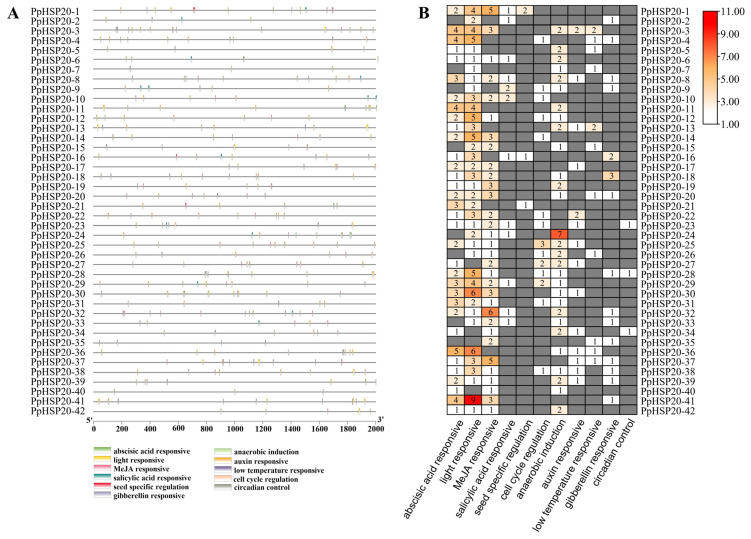
Analysis of the *cis*-elements in the *PpHSP20s* promoters. (**A**) The different types of *cis*-elements are shown in the promoter region of each *PpHSP20* in different colors. (**B**) The number of each *cis*-acting element in the promoter regions of each *PpHSP20*. Coloring represents the number of elements (small: white; large: red), gray indicate zero.

**Figure 5 ijms-23-10849-f005:**
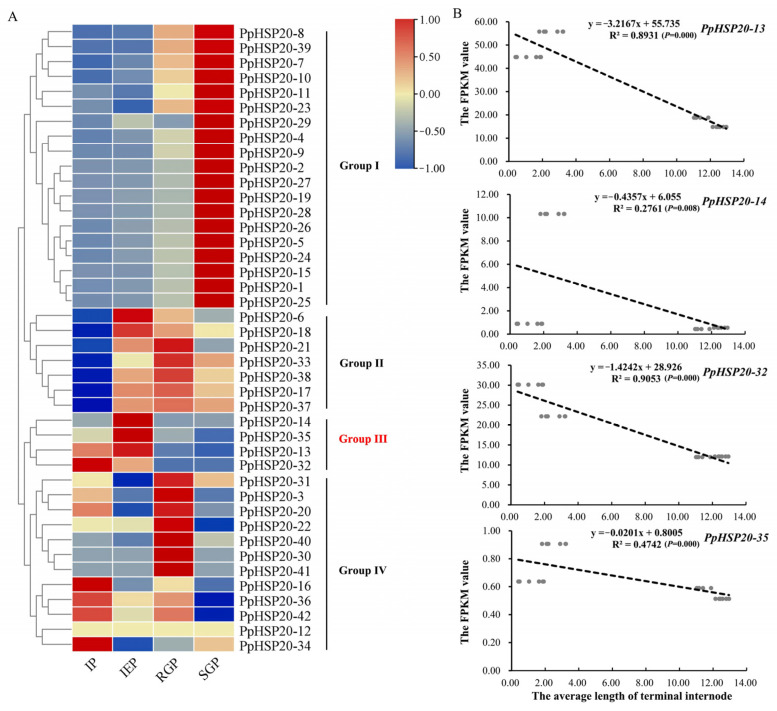
Expression levels of *PpHSP20s* during shoot elongation in ‘Zhongyoutao 14’ and correlation with the average terminal internode length (TIL). (**A**) Transcriptome data was used to measure the expression level of the *PpHSP20s*. The growth stages were the IP (initial period), IEP (initial elongation period), RGP (rapid growth period), and SGP (stable growth period), corresponding to four key growth stages during temperature-sensitive peach shoot development. Four groups of transcriptional patterns were classified. ‘Group III’ is highlighted with red, showing a negative trend between the internode length and the expression levels of *PpHSP20s* in Group III. (**B**) Correlation between the TIL and the expression level of genes in Group III. Black dotted line represents linear trend. Grey dots represent the internode length and the FPKM value. The average length was calculated by measuring six terminal internodes at each stage, n = 6. FPKM, fragments per kilobase of transcript per million mapped reads.

**Figure 6 ijms-23-10849-f006:**
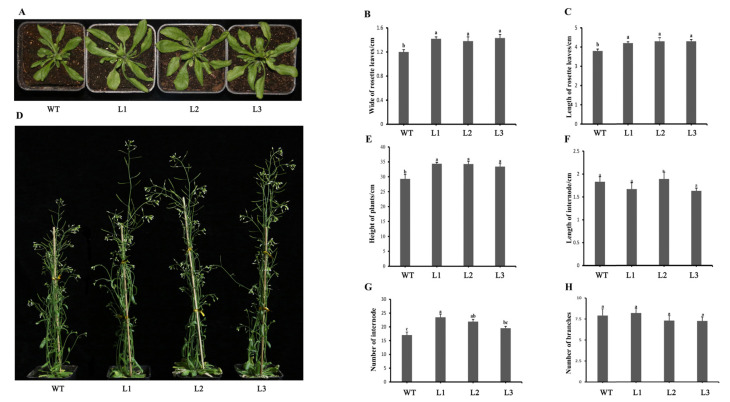
Phenotypic analysis of transgenic Arabidopsis overexpressing PpHSP20-32. (**A**) Phenotypes of T_2_ transgenic plants from three lines over-expressing PpHSP20-32 after cultivation for two weeks. The length (**B**) and width (**C**) of the rosette leaves after cultivation for two weeks. (**D**) Phenotypes of T_2_ transgenic plants after cultivation for four weeks. The plant height (**E**), internode length (**F**), number of nodes (**G**), and branches (**H**) after cultivation for four weeks. Different letters indicate significant differences within treatments by ANOVA (*p* < 0.05).

**Figure 7 ijms-23-10849-f007:**
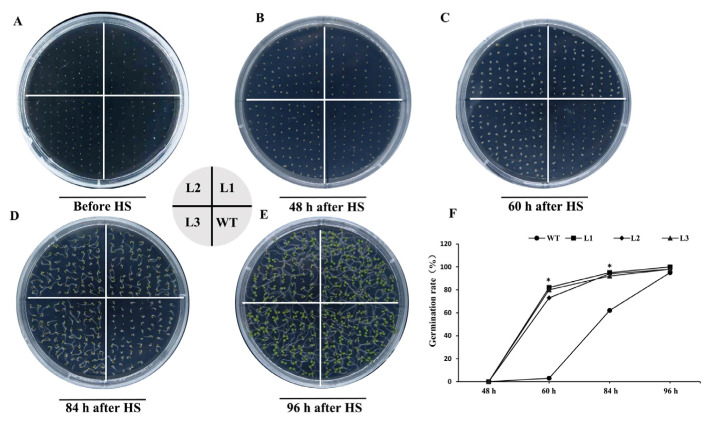
Thermotolerance of the PpHSP20-32-OE lines. (**A**) Seeds of wild type (WT) and the PpHSP20-32-OE lines (L1, L2, and L3) were treated at 46 °C for 30 min. Photographs were taken before HS treatment. (**B**–**E**) Photographs were taken after 48 h, 60 h, 80 h, and 96 h at 25 °C. (**F**) Germination rates among wild-type and PpHSP20-32-OE lines transgenic plants after HS treatment. The number of germinated plants was counted at different times after HS treatment. For three replications, more than 50 seedlings were used for each line (*t*-test significant at *p* < 0.05).

**Table 1 ijms-23-10849-t001:** Predicted protein characteristics of the PpHSP family members.

Gene ID	Gene Name	Amino Acid Number	Molecular Weight/Da	Isoelectric Point	Instability Index	Aliphatic Index	Average of Hydropathicity
Prupe.1G237800	*PpHSP20-1*	237	26,812.57	9.13	53.71	57.59	−0.802
Prupe.1G407100	*PpHSP20-2*	162	18,223.84	6.64	53.2	85.99	−0.54
Prupe.1G407200	*PpHSP20-3*	214	22,898.39	5.78	45.85	65.09	−0.86
Prupe.1G407300	*PpHSP20-4*	156	17,514.17	5.58	39.38	80.51	−0.439
Prupe.1G586200	*PpHSP20-5*	202	22,714.25	9.12	34.67	90.4	−0.356
Prupe.2G065600	*PpHSP20-6*	438	47,399.43	9.18	41.9	66.28	−0.828
Prupe.2G243400	*PpHSP20-7*	165	18,345.7	6.21	48.88	76.12	−0.621
Prupe.2G243800	*PpHSP20-8*	154	17,381.68	5.81	55.45	81.04	−0.671
Prupe.2G317700	*PpHSP20-9*	223	24,801.18	7.75	47.95	85.61	−0.533
Prupe.3G017400	*PpHSP20-10*	143	16,084.37	6.84	42.01	85.17	−0.45
Prupe.3G034800	*PpHSP20-11*	161	17,783.21	6.16	42.95	82.92	−0.463
Prupe.3G085200	*PpHSP20-12*	234	26,468.86	6.87	55.03	86.2	−0.672
Prupe.3G108500	*PpHSP20-13*	245	27,393.14	9.26	58.68	75.27	−0.469
Prupe.4G023100	*PpHSP20-14*	133	15,541.7	5.52	49.18	89.4	−0.353
Prupe.4G125800	*PpHSP20-15*	160	18,417.87	7.13	37.8	73.06	−0.832
Prupe.4G201100	*PpHSP20-16*	219	25,002.23	6.91	50.18	73.42	−0.624
Prupe.5G004500	*PpHSP20-17*	399	44,140.85	6.35	31.61	75.21	−0.619
Prupe.5G006000	*PpHSP20-18*	138	14,813.27	8.54	41.03	100.14	0.082
Prupe.5G071200	*PpHSP20-19*	209	23,640.61	5.46	70.11	74.07	−0.626
Prupe.5G186700	*PpHSP20-20*	327	37,390.03	5.13	55.85	81.07	−0.488
Prupe.5G186800	*PpHSP20-21*	233	25,847.96	5.61	44.56	78.58	−0.609
Prupe.5G242900	*PpHSP20-22*	214	23,777.01	9.26	28.11	75.09	−0.575
Prupe.6G008800	*PpHSP20-23*	143	16,155.26	4.69	67.62	83.85	−0.298
Prupe.6G065900	*PpHSP20-24*	158	18,076.55	7.91	50.58	74.62	−0.666
Prupe.6G066100	*PpHSP20-25*	158	18,105.5	5.83	49.72	73.99	−0.683
Prupe.6G066200	*PpHSP20-26*	158	17,961.49	6.76	48.21	80.13	−0.555
Prupe.6G066300	*PpHSP20-27*	161	18,427.82	5.57	49.59	73.23	−0.671
Prupe.6G066400	*PpHSP20-28*	158	18,222.61	6.34	51.41	71.52	−0.697
Prupe.6G066500	*PpHSP20-29*	158	18,109.53	5.57	55.98	77.09	−0.613
Prupe.6G066600	*PpHSP20-30*	87	9408.79	9.1	57.83	74.02	−0.299
Prupe.6G151500	*PpHSP20-31*	331	36,970.14	9.32	43.37	63.08	−0.871
Prupe.6G266500	*PpHSP20-32*	153	17,931.3	6.45	49.89	65.56	−0.746
Prupe.6G328400	*PpHSP20-33*	204	22,150.2	5.1	29.97	79.75	−0.317
Prupe.7G008300	*PpHSP20-34*	174	19,246.36	9.33	39.03	86.32	−0.243
Prupe.7G008400	*PpHSP20-35*	160	18,391.29	9.28	43.56	85.81	−0.296
Prupe.7G008500	*PpHSP20-36*	147	16,918.68	9.44	32.95	88.78	−0.276
Prupe.7G053900	*PpHSP20-37*	428	48,041.14	5.88	43.93	77.15	−0.42
Prupe.7G187900	*PpHSP20-38*	572	62,944.2	4.93	48.98	60.28	−0.957
Prupe.8G000400	*PpHSP20-39*	200	22,775.12	5.91	46.96	97.4	−0.442
Prupe.8G031300	*PpHSP20-40*	485	54,400.11	5.36	50.45	72.76	−0.911
Prupe.8G046200	*PpHSP20-41*	163	18,241.31	5.62	48.21	67.48	−0.799
Prupe.8G182600	*PpHSP20-42*	919	100,089.87	4.59	43.82	74.11	−0.633

**Table 2 ijms-23-10849-t002:** The secondary structure analysis and subcellular localization prediction of PpHSP proteins.

Protein ID	Alpha Helix	Extended Strand	Beta Turn	Random Coil
PpHSP20-1	51	34	6	146
PpHSP20-2	56	34	12	60
PpHSP20-3	60	35	17	102
PpHSP20-4	49	36	10	61
PpHSP20-5	78	29	12	83
PpHSP20-6	154	36	8	240
PpHSP20-7	34	29	9	93
PpHSP20-8	32	32	7	83
PpHSP20-9	32	46	10	135
PpHSP20-10	32	31	10	70
PpHSP20-11	50	33	9	69
PpHSP20-12	84	33	13	104
PpHSP20-13	51	38	8	148
PpHSP20-14	28	31	7	67
PpHSP20-15	33	38	6	83
PpHSP20-16	62	46	18	93
PpHSP20-17	103	48	18	230
PpHSP20-18	14	44	8	72
PpHSP20-19	49	30	11	119
PpHSP20-20	99	53	28	147
PpHSP20-21	57	36	14	126
PpHSP20-22	56	49	12	97
PpHSP20-23	15	38	10	80
PpHSP20-24	24	24	10	91
PpHSP20-25	26	34	8	90
PpHSP20-26	29	31	8	90
PpHSP20-27	28	28	8	97
PpHSP20-28	29	27	7	95
PpHSP20-29	30	30	9	89
PpHSP20-30	13	14	3	57
PpHSP20-31	143	20	8	160
PpHSP20-32	31	35	10	77
PpHSP20-33	21	53	13	117
PpHSP20-34	60	30	30	30
PpHSP20-35	48	34	12	66
PpHSP20-36	56	29	12	50
PpHSP20-37	85	84	27	232
PpHSP20-38	144	67	23	338
PpHSP20-39	76	31	9	84
PpHSP20-40	202	61	30	192
PpHSP20-41	42	29	8	84
PpHSP20-42	312	130	51	426

## Data Availability

Not applicable.

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
