# Peer review of "Phylogenetic and Transcriptional Analyses of the HSP20 Gene Family in Peach Revealed That PpHSP20-32 Is Involved in Plant Height and Heat Tolerance"

_ijms, 2022, doi:10.3390/ijms231810849_

Round 1
Reviewer 1 Report
The paper entitled "Phylogenetic and Transcriptional Analysis of the HSP20 gene family in peach revealed that PpHSP20-32 is involved in plant height and heat tolerance" is a well-written manuscript that analyzes the HSP20 genes in peach, which is novel. A simple literature review resulted in a small number of publications on this topic, and the authors have expertise.
The introduction section is straight to the point and states the scope of the research. Authors are encouraged to highlight why peach was selected and if there is a scientific background behind this selection.
The materials and methods section is straightforward and allows the methodology's repeatability. However, in Lines 309-312, the authors should (1) rephrase the first sentence, (2) mention why this semi-dwarf cultivar was selected and also refer to this in the discussion section (would be different results in other cultivars?), (3) add the word critical growth stages, and (4) mention the range of an important characteristic for the IP phase (e.g., leaves, height etc.). In addition, it is suggested that the authors present an estimate (possibly in discussion) of what would happen if the heat stress treatment lasted 1 h or more.
The results and discussion sections are sound and clear. Please add more notes on the regular management of this cultivar in Line 173.
Reviewer 2 Report
This study phylogenetically and transcriptionally analyzed 42 HSP20 genes in peach and evaluated the role of PpHSP20-32 in plant performance using Arabidopsis thaliana. Overall, this is an interesting study, but some weaknesses should be addressed before further consideration. Firstly, English should be further polished (see my below specific comments). Secondly, it is a lack of objectives/scientific questions and hypotheses in this study. Thirdly, the discussion is so weak, and some parts look like an introduction rather than a discussion. Lastly, there is no conclusion section in this study.
Round 2
Reviewer 2 Report
Overall, the authors address most of the comments. However, there are several comments that the authors may not accurately address.
1)To be honest, L68-86 are not objectives and hypotheses. Please modify them.
2) Fig. 5B. The authors may not completely understand my previous point. It is more statistically powerful to use all measure points to make a regression compared to using the average at each stage.
3). L410-413. Unclear!
